# Role of Nitric Oxide in the Antidepressant Effect of an Aqueous Extract of *Punica granatum* L.: Effects on GSH/GSSG Ratio and Lipoperoxidation in Adult Male Swiss Webster Mice

**DOI:** 10.3390/ijms262110255

**Published:** 2025-10-22

**Authors:** Nancy Cervantes-Anaya, Alexandere Aedo-Torrado, Erika Estrada-Camarena, Verónica Pérez de la Cruz, Daniela Ramírez Ortega, María Eva González-Trujano, Carolina López-Rubalcava

**Affiliations:** 1Laboratorio de Neuropsicofarmacología, Dirección de Neurociencias, Instituto Nacional de Psiquiatría Ramón de la Fuente Muñiz, Calzada México-Xochimilco 101, Ciudad de México 14370, Mexico; 2Departamento de Farmacobiología, Centro de Investigación y de Estudios Avanzados del IPN, Ciudad de México 14330, Mexico; 3Laboratorio de Neurobioquímica y Conducta, Instituto Nacional de Neurología y Neurocirugía Manuel Velasco Suárez, Secretaria de Salud (SSA), Ciudad de México 14269, Mexico; 4Laboratorio de Neurofarmacología de Productos Naturales, Dirección de Investigaciones en Neurociencias, Instituto Nacional de Psiquiatría Ramón de la Fuente Muñiz, Ciudad de México 14370, Mexico

**Keywords:** *Punica granatum*, nitric oxide, depression, 7-nitroindazole (7-NI), NG-nitro-L-arginine methyl ester (L-NAME)

## Abstract

Depression is a prevalent psychiatric disorder in which oxidative stress and nitric oxide (NO) signaling have been implicated. Natural compounds such as *Punica granatum* have shown potential antidepressant effects, but their mechanisms remain unclear. This study aimed to evaluate the antidepressant-like effects of an aqueous extract of *P. granatum* in male Swiss Webster mice and to explore the possible involvement of NO-related system. Acute administration of *P. granatum* (0.25–2 mg/kg, i.p.) was tested in the forced swim and tail suspension tests. The interaction with NO signaling was examined through co-administration with an NO donor (sodium nitroprusside, SNP) and NOS inhibitors (NG-nitro-L-arginine methyl ester, L-NAME and 7-nitroindazole, 7-NI). Biochemical markers of oxidative stress (lipid peroxidation and GSH/GSSG ratio) were also assessed. *P. granatum* significantly reduced immobility and increased swimming behavior, consistent with an antidepressant-like effect. SNP, L-NAME, and 7-NI induced pro-depressant effects, which were prevented by *P. granatum* co-administration. Treatment also attenuated oxidative stress markers in the hippocampus and prefrontal cortex. These findings suggest that the antidepressant-like effects of *P. granatum* may involve interactions with NO signaling, although this interpretation remains indirect, as specific NO pathway indices were not measured. Acute *P. granatum* administration exerts antidepressant-like and antioxidant effects in male mice. The results support its potential as a natural candidate for depression treatment, particularly in conditions associated with oxidative imbalance and possible NO dysregulation. Future studies should confirm these mechanisms using direct molecular assessments and include female cohorts.

## 1. Introduction

Depression is a common and disabling psychiatric disorder, with the World Health Organization (WHO) estimating it to be the second leading cause of global disease burden. It affects over 25% of women and 12% of men [1]. About 30% of depressed patients do not respond to current medications, and 70% do not achieve complete remission [2,3]. Furthermore, approximately 28% of patients discontinue antidepressants within the first month, increasing to 44% by the third month [4]. These challenges underscore the urgent need for new therapeutic targets and alternative treatments [5,6].

Oxidative stress plays a significant role in the pathophysiology of depression. The brain is especially vulnerable to oxidative damage due to its high content of polyunsaturated fatty acids and its relatively low antioxidant capacity, making it highly susceptible to injury from free radicals that trigger lipid peroxidation (LPO). Elevated iron levels in the brain can further exacerbate oxidative stress and promote cellular damage [7,8]. Patients with depression often exhibit reduced antioxidant defenses [9,10], characterized by lower levels of vitamin E, coenzyme Q10, tryptophan, tyrosine, estrogens, zinc, and albumin, along with a reduced total antioxidant capacity [11,12]. This imbalance can be exacerbated by hyperactivity of the Hypothalamic–Pituitary–Adrenal (HPA) axis, a common feature in individuals with depression under chronic psychosocial stress [13,14].

Among endogenous antioxidants, glutathione (GSH) plays a central role in neutralizing reactive oxygen species (ROS) and maintaining redox homeostasis. While GSH is normally found in its reduced form, its oxidized form (GSSG) increases under oxidative conditions, making the GSSG/GSH ratio a widely used marker of oxidative stress [8,15,16,17].

In addition to oxidative imbalance, nitric oxide (NO) signaling has been implicated in depression. NO is a gaseous neurotransmitter synthesized from L-arginine by nitric oxide synthases (NOS): neuronal (nNOS), inducible (iNOS), and endothelial (eNOS). It regulates the HPA axis and modulates several neurotransmitter systems, including serotonin (5-HT), dopamine, noradrenaline, and glutamate [10,12,18,19]. Notably, NO has a dual role in neurogenesis and may influence antidepressant efficacy, particularly of selective serotonin reuptake inhibitors (SSRIs). In animal models, NOS inhibitors such as NG-nitro-L-arginine methyl ester (L-NAME, a non-selective agent) and 7-nitroindazole (7-NI, a specific inhibitor of nNOS) produce antidepressant-like effects by reducing immobility in behavioral despair tests and enhancing monoaminergic transmission in the hippocampus [12,14,19,20].

Pomegranate (*Punica granatum* L.) has demonstrated antidepressant-like effects in animal models, comparable to conventional antidepressants such as fluoxetine and imipramine [9,11,16,17]. These effects have been observed following subacute and chronic administration [13,15,18,19]. Natural compounds with antioxidant properties, including ellagitannins and polyphenols present in *P. granatum*, show promise as alternative treatments for depression [3,6,7,20].

Moreover, in vitro studies have shown that *P. granatum* exerts anti-inflammatory and antioxidant effects by inhibiting NO production and suppressing iNOS expression in LPS-stimulated RAW 264.7 macrophage cell lines. These actions are attributed to hydrolyzable tannins such as punicalagin, punicalin, strictin A, and granatin B [21]. However, whether NO inhibition contributes to the antidepressant-like effects of *P. granatum* in vivo remains unclear.

The present study aimed to evaluate the role of NO in the antidepressant effects of *P. granatum,* focusing on its impact on depressive-like behaviors and oxidative stress markers, including the GSH/GSSG ratio and lipid peroxidation (LPO) levels, in specific brain regions under both stress and non-stress conditions. We hypothesized that *P. granatum* exerts its antidepressant-like effects, at least in part, by modulating NO signaling and restoring oxidative balance. In addition, to better characterize the pharmacological profile of *P. granatum*, we determined the effective dose 50 (ED50) for its antidepressant-like effects in the forced swim test (FST).

## 2. Results

### 2.1. Chromatographic Profile of P. granatum Aqueous Extract

Figure 1 displays the chromatographic profile derived from the aqueous extract of P. granatum, confirming the presence of punicalagin isomers (α and β) and derivatives of ellagic acid.

### 2.2. Effect of P. granatum on Tail Suspension Test (TST) and Forced Swimming Test (FST)

The activity test was conducted to eliminate any nonspecific effects of the drugs under investigation. During this test, it was observed that a dose of 2 mg/kg of the extract impacted general activity. Consequently, this dose was not used in subsequent behavioral tests.

In the TST, a significant decrease in immobility time (F:_4, 45_ = 4.15; *p* < 0.006; η^2^ = 0.269) was observed for *P. granatum* at a dose of 1.0 mg/kg (Figure 2A). In FST, a significant decrease (F:_4, 45_ = 5.93; *p* < 0.006; η^2^ = 0.345) in immobility behavior was observed for both the 0.5 mg/kg dose and the 1.0 mg/kg dose (*p* ≤ 0.01) compared to the control (Figure 2B). This decrease is due to an increase in swimming behavior that did not reach statistical significance (F:_4, 45_ = 1.81; ns; η^2^ = 0.138) (Figure 2C). Regarding climbing behavior, no significant differences (F:_4, 45_ = 1.53; ns; η^2^ = 0.12) were observed compared to the control group (Figure 2D). The dose–response curve for *P. granatum* revealed a significant antidepressant-like effect, with an estimated ED50 of 0.45 mg/kg in the FST.

### 2.3. Effect of Sodium Nitroprusside (SNP), NO Donor, on TST and FST

In the TST, a significant increase in immobility time (F:_3, 35_ = 7.48; *p =* 0.0005; η^2^ = 0.39) was observed only for the 1.0 mg/kg dose (*p* ≤ 0.01) compared to the control group (Figure 3A). This result coincides with that observed in the FST since an increase in immobility behavior (F:_3, 36_ = 5.83; *p =* 0.002; η^2^ = 0.32) was observed at the 1.0 mg/kg dose of SNP (Figure 3B), at the expense of decreases in swimming (F:_3, 36_ = 2.27; *p =* 0.09; η^2^ = 0.15) and climbing behavior (F:_3, 36_ = 6.06; *p =* 0.002; η^2^ = 0.33) compared to the control group (Figure 3C,D).

### 2.4. Effect of L-NAME, NOS Inhibitor, on TST and FST

In the TST, L-NAME at doses of 2.5, 5.0, 10, 20, and 40 mg/kg produced no significant changes in immobility time (F:_5, 54_ = 0.45; ns; η^2^ = 0.040) compared to the control group (Figure 4A). In contrast, in the FST, L-NAME significantly decreased immobility (F:_5, 54_ = 26.50; *p* < 0.0001; η^2^ = 0.71) at doses ranging from 5.0 to 40 mg/kg (*p* ≤ 0.01) with respect to the control; whereas the lowest doses (2.5 mg/kg) significantly increased immobility (*p* ≤ 0.05) (Figure 4B). A significant increase in swimming behavior accompanied the reduction in immobility (F:_5, 54_ = 8.92; *p* < 0.0001; η^2^ = 0.45) at doses of 5, 10, 20, and 40 mg/kg (*p* ≤ 0.05) compared to the control group (Figure 4C). A significant reduction in climbing behavior (F:_5, 54_ = 3.37; *p* = 0.01; η^2^ = 0.23) was observed with the L-NAME dose of 25 mg/kg compared with the control group (Figure 4D).

### 2.5. Effect of 7-NI, a Selective Inhibitor of nNOS, on the TST and FST

In the TST, 7-NI produced no significant changes in immobility time (F:_3, 34_ = 2.20; ns; η^2^ = 0.16) at any of the tested doses (Figure 5A). In contrast, in the FST, immobility time increased significantly (F:_3, 35_ = 5.23, *p =* 0.004; η^2^ = 0.30) at the highest dose of 40 mg/kg (*p* ≤ 0.01) compared to control group (Figure 5B), promoting a significant reduction in climbing behavior (F:_3, 35_ = 5.001; *p* = 0.005; η^2^ = 0.30) compared to the control group (*p* ≤ 0.01) An increase on swimming behavior (F:_3, 35_ = 4.40; *p =* 0.01; η^2^ = 0.27) was observed that was significant at dose of 80 mg/kg compared with control group (*p =* 0.05) (Figure 5C,D).

### 2.6. Determination of the Effect of the Combination of P. granatum Plus SNP in the TST and FST

In the TST, *P. granatum* (1 mg/kg) significantly reduced the immobility compared to the control group (*p* < 0.05), whereas SNP (1 mg/kg) produced the opposite effect, significantly increasing immobility (*p* < 0.005). Interestingly, co-administration of *P. granatum* and SNP prevented an increase in immobility induced by SNP alone (F:_5, 53_ = 7.10; *p* < 0.0001; η^2^ = 0.40) (Figure 6A).

In the FST (Figure 6B), *P. granatum* (1 mg/kg) markedly decreased immobility behavior (*p* < 0.001), while SPN (1.0 mg/kg) increased it (*p* < 0.001) compared to the control group. The combination of *P. granatum* plus SNP did not differ from the control group; however, it abolishes the individual effects of both *P. granatum* (*p* ≤ 0.001) and SNP (*p* ≤ 0.001) alone (F:_5, 54_= 43.22; *p* < 0.0001; η^2^ =0.80) (Figure 6B). Regarding swimming behavior (Figure 6C), the *P. granatum* group exhibited a significant increase compared to the combination group (*p* ≤ 0.01). In contrast, climbing behavior (Figure 6D) showed a significant difference in the SNP group (*p* < 0.001) compared to the control group. One-way ANOVA yielded the following values for swimming (F:_5, 54_ = 11.02; *p* < 0.0001; η^2^ = 0.50) and for climbing (F:_5, 54_ = 7.83; *p* < 0.0001; η^2^ = 42).

### 2.7. Effect of the Combination of P. granatum and L-NAME in the TST and FST

In the TST, *P. granatum* (1 mg/kg) significantly reduced immobility time compared to the control group (*p* < 0.05). This effect was completely abolished when *P. granatum* was co-administered with L-NAME (2.5 mg/kg), a dose that, when administered alone, did not significantly affect immobility (F:_3, 36_ = 2.93; *p =* 0.04; η^2^ = 0.19) (Figure 7A). In the FST (Figure 7B), *P. granatum* (1 mg/kg) reduced immobility behavior (*p* ≤ 0.001), while L-NAME (2.5 mg/kg) significantly increased it (*p* ≤ 0.001). When combined, L-NAME partially reversed the antidepressant-like effect of *P. granatum* (*p* ≤ 0.001) and completely blocked its own immobility-enhancing effect (*p* ≤ 0.001). Furthermore, the combination resulted in a reduction in immobility compared to the control group (F:_3, 36_ = 48.22; *p* < 0.0001; η^2^ = 0.80) (Figure 7B).

An increase in swimming behavior accompanied the decrease in immobility behavior (F:_3, 36_ = 25.51; *p* < 0.0001; η^2^ = 0.68) (Figure 7C) in the group of *P. granatum* and the combined group when compared with the control group (*p* < 0.001) and the L-NAME group (*p* < 0.001). The increase in immobility behavior observed when comparing the combined group with the *P. granatum* group appears to be due to a significant decrease in climbing behavior in the combined-dose group (F:_3, 36_ = 7.84; *p =* 0.004; η^2^ = 0.39). (Figure 7D).

### 2.8. Effect of the Combination of P. granatum and 7-NI in the TST and FST

In the TST, *P. granatum* significantly reduced immobility compared to the control group (*p* ≤ 0.01). This effect was abolished entirely when it was co-administered with 7-NI (*p* ≤ 0.01) relative to the *P. granatum* group. Administration of 7-NI alone did not significantly alter immobility (Figure 8A). One-way ANOVA yielded the following value (F:_3, 34_ = 7.27; *p =* 0.0007; η^2^ = 0.39). In FST (Figure 8B), *P. granatum* (1 mg/kg) again produced a marked reduction in immobility (*p* ≤ 0.001), while 7-NI significantly increased it. When both compounds were administered together, the antidepressant-like effect of *P. granatum* was entirely blocked (*p* < 0.001), and no additional effects beyond those induced by 7-NI were observed (Figure 8B). We also observed a significant decrease (*p* < 0.01) in both swimming (Figure 8C) and climbing behavior (Figure 8C) when compared with the group that received only *P. granatum*. One-way ANOVA yielded the following values for immobility (F:_3, 36_ = 20.55; *p* < 0.0001; η^2^ = 0.63); for swimming (F:_3, 36_ = 8.72; *p* = 0.0002; η^2^ = 0.42); and for climbing (F:_3, 36_ = 6.73; *p* = 0.001; η^2^ = 0.35).

### 2.9. Effect of P. granatum on Lipid Peroxidation in Brain and Serum in Mice Exposed, or Not, to FST

LPO is a well-established biomarker of oxidative stress; here, we evaluated the effect of *P. granatum* on redox environment using this marker. As shown in Figure 9, exposure to the FST significantly increased the LPO in the frontal cortex (F:_3, 16_ = 8.17; *p* < 0.001; η^2^ = 0.60) (Figure 9A), in the hippocampus (F:_3, 16_ = 6.30; *p* = 0.005; η^2^ = 0.540) (Figure 9B), amygdala (F:_3, 16_ = 7.78; *p* = 0.002; η^2^ = 0.59) (Figure 9C), but not in the brain stem (ns; Figure 9D) compared with saline-no FST group.

*P. granatum* did not alter basal LPO level in mice not subjected to the FST in any brain region assessed (Figure 9A–D). However, in mice subjected to FST in all areas evaluated, *P. granatum* administration prevented the FST-induced increase in LPO (*p* ≤ 0.02) (Figure 9). Finally, serum LPO levels did not differ significantly among any of the treatment groups (Figure 9E). Two-way ANOVA test yielded the following values for the cortex factor A (behavioral test) F:_1, 16_ = 3.93, *p =* 0.063, η^2^ = 0.09; factor B (treatment) F:_1, 16_ = 3.84, *p =* 0.067, η^2^ = 0.09; and the interaction AXB F:_1, 16_ = 16.54; *p* < 0.0001; η^2^ = 0.40. For hippocampus, factor A F:_1, 16_ = 9.689; *p =* 0.007; η^2^ = 0.27; Factor B F:_1, 16_ = 6.439; *p =* 0.02; η^2^ = 0.18; and the interaction AXB F:_1, 16_ = 2.77; ns; η^2^ = 0.07. For amygdala factor A F:_1, 16_ = 14.67, *p =* 0.001, η^2^ = 0.37; factor B F:_1, 16_ = 3.036, ns, η^2^ = 0.07; and the interaction AXB F:_1, 16_ = 5.64, *p =* 0.03, η^2^ = 0.14. For brain stem factor A F:_1, 16_ = 0.758, ns, η^2^ = 0.021; factor B F:_1, 16_ = 10.50, *p =* 0.006, η^2^ = 0.28; and the interaction AXB F:_1, 16_ = 8.475, *p =* 0.01, η^2^ = 0.29.

### 2.10. Effect of P. granatum on Redox Status in Mice Brains

To assess the redox status in mice exposed, or not, to FST, levels of reduced glutathione (GSH) and oxidized glutathione (GSSG) were determined in the cortex, hippocampus, brainstem, and serum (Figure 10). In the cortex (Figure 10A), *P. granatum* administration significantly increased the ratio GSH/GSSG in non-swimming mice compared to the swimming control group (*p* ≤ 0.01). In mice exposed to the FST, no significant changes in the GSH/GSSG ratio were observed in either the saline or *P. granatum* groups. Regarding the hippocampus (Figure 10B), *P. granatum* treatment significantly increased the GSH/GSSG ratio in non-swimming mice compared to the non-swimming control group (*p* ≤ 0.01). Similarly to the cortex, no significant changes were observed in either saline or *P. granatum*-treated mice subjected to the FST. For the brain stem and serum (Figure 10C,D), the GSH/GSSG ratio followed a pattern similar to that observed in the cortex and hippocampus; however, these differences were not statistically significant in any of the treatment groups. The two-way ANOVA test yielded the following values for cortex factor A F:_1, 16_ = 5.93, *p =* 0.027, η^2^ = 0.19; factor B F:_1, 16_ = 5.94, *p =* 0.027, η^2^ = 0.19; and the interaction AXB F:_1, 16_ = 2.57, ns, η^2^ = 0.08. For hippocampus factor A F:_1, 16_ = 5.61, *p =* 0.03; η^2^ = 0.15; factor B F:_1, 16_ = 13.46, *p =* 0.002; η^2^ = 0.36.

Pearson correlation between the variables measured in the FST, LPO, and GSH/GSSG showed that immobility behavior exhibits a medium positive correlation with LPO in the frontal cortex (r = 0.86; *p* < 0.05), hippocampus (r = 0.56; *p* = 0.05), and amygdala (r = 0.68; *p* < 0.05) (Figure 11). In contrast, swimming behavior displays a slight positive correlation with GSH/GSSG in cortex (r = 0.21) and hippocampus (0.38). Climbing behavior, however, shows no association with either LPO or oxidative stress. The present data reinforce the suggestion that *P. granatum* mitigates the effects of LPO induced by the FST and reduces oxidative stress.

### 2.11. Effect of P. granatum and the Different Study Treatments in the Locomotor Activity Test

The effects of *P. granatum*, SNP, L-NAME, and 7-NI, both individually and in combination, are presented in Figure 12. The results indicate that only the higher doses of *P. granatum* (2 mg/kg) and SNP (4 mg/kg) lead to a decrease in locomotor activity. Additionally, *P. granatum* (1.0 mg/kg) combined with SNP (1.0 mg/kg) also reduce general activity in mice. While these treatments impacted locomotor activity, they did not affect the animals’ performance in the forced swimming test; in fact, the same treatments were associated with an increase in active behaviors, as shown in Figure 6. Similar effects were observed with the combined treatment of *P. granatum* (1 mg/kg) and 7-NI (40 mg/kg). Table 1 presents the results from the One-Way ANOVA test conducted on the different treatments in the locomotor activity test.

## 3. Discussion

The TST and FST are widely used methods to evaluate potential antidepressant drugs [22]. Our results demonstrated that *P. granatum* induced antidepressant-like effects in both tests, as evidenced by a reduction in immobility behavior, consistent with previous findings in female rats [11,13,15,23]. *P. granatum* appears to mediate its antidepressant-like effects through the serotonergic system, selectively increasing swimming behavior in the FST [24]. Furthermore, earlier studies have shown that degeneration of serotonergic presynaptic neurons blocks the antidepressant-like effect of *P. granatum* [13], supporting the involvement of this system in those effects.

A novel aspect of our study was the evaluation of the role of NO in the antidepressant-like effects of *P. granatum*. We found that SNP, an NO donor, produced a prodepressant-like effect at 1.0 mg/kg, increasing immobility in both the TST and the FST, while reducing swimming and climbing in the latter test. Similar prodepressant effects have been reported with L-arginine and SNP in the FST and socially isolated mice, while increasing nitric oxide levels in the hippocampus and frontal cortex. This suggests the role of nitric oxide in the regulation of depressive-like behaviors [25,26,27].

Some NO donors can exhibit dose-dependent biphasic effects, suggesting that NO may function as a metabolic regulator with adaptogenic properties. This could enhance adaptation to environmental factors and help mitigate potential damage caused by those factors [28,29,30]. Importantly, our study focused on depressive-like behavior, unlike most previous research that examined anxiety-like behavior [30,31,32].

In contrast, the NOS inhibitor L-NAME showed a dose-dependent effect. Low doses (2.5 mg/kg) produced a mild prodepressant-like profile, characterized by increased immobility behavior and decreased active behaviors. In contrast, higher doses (5–40 mg/kg) resulted in significant antidepressant-like effects, indicated by decreased immobility and an increase in swimming behavior primarily. This hormetic effect suggests that there is a threshold of NO inhibition required to trigger antidepressant-like behavior. A similar effect was observed in the FST with the selective nNOS inhibitor 7-NI. Specifically, a dose of 40 mg/kg increased immobility, while doses of 80 mg/kg reduced it. This suggests a dose-dependent effect on the monoaminergic pathways. The current data support the role of nitric oxide in regulating depressive-like behaviors.

Previous studies indicate that inhibiting NO production is consistently associated with anxiolytic and antidepressant effects [33], while an increase in NO favors prodepressant effects and oxidative stress in the brain [27]. For example, blocking NO can reduce serotonin depletion in the prefrontal cortex, producing effects similar to those of imipramine [34]. Moreover, fluoxetine has been shown to downregulate nNOS in the hippocampus [35]. The present study aligns with these findings, as a positive correlation was observed between immobility behavior and LPO in the hippocampus, amygdala, and cortex, indicating an increase in oxidative stress induced by FST.

Conversely, both selective and non-selective NOS inhibitors, such as L-NAME and 7-NI, have been found to diminish ACTH response to stress. Given that nNOS is enhanced by corticosterone during stress, its inhibition may interfere with NO’s role in activating the HPA axis and contribute to stress-related comorbidities [36]. Therefore, it is possible that at high doses, NOS inhibitors used here contribute to reducing stress and promote antidepressant-like actions. Supporting this notion, other studies have shown that NO inhibitors can have dual effects. For instance, L-NAME has been found to produce anxiogenic effects that depend on the dose and the level of stress [37]. On the other hand, in rats subjected to restraint stress, L-NAME prevents anxiety- and depression-like behaviors. This suggests that NO can have a hormetic effect on stress adaptation, where low doses are protective while high doses are detrimental [28].

Results from the present studies support the idea that NO exerts a dual role in mood regulation, potentially acting as an adaptogen depending on context, stress level, and concentration. The differential effects observed across doses and inhibitors highlight the complexity of NO’s involvement in depressive-like conditions.

In determining the effect of combined doses of *P. granatum* with SNP, previous findings suggest that the nitrergic system plays a role in both the induction and inhibition of depressive-like behaviors [19,38]. Based on this, the present study examined how *P. granatum* may interact with this system. When combining an effective dose of SNP (1.0 mg/kg), which produces a prodepressant-like effect, with an effective dose of *P. granatum* (1.0 mg/kg), the prodepressant-like effect of SNP was only partially blocked. However, the combined treatment did not reproduce the full antidepressant effect observed when *P. granatum* was administered alone. This suggests that *P. granatum*’s antidepressant action on the nitrergic system likely involves modulation of NO synthase activity. The addition of NO-releasing prodrugs such as SNP, which bypasses the enzyme system, may override the extract’s capacity to maintain its antidepressant efficacy, again pointing to a stress-like simulation by NO that exceeds the adaptogenic buffering capacity of *P. granatum* [28]. Similar findings have been reported with L-arginine on the antidepressant-like actions of quercetin, a potent polyphenol [25].

Although no prior studies have examined this specific drug combination, *P. granatum*’s individual properties support the hypothesis that its serotonergic antidepressant effects may be antagonized by the excess NO produced by SNP [13]. Another explanation could be that *P. granatum* reduces NO production by downregulating the expression of nNOS and iNOS, while SNP, acting independently of these enzymes, continues to increase NO availability [21]. This may explain why the residual antidepressant-like effect observed with *P. granatum* is diminished in the presence of SNP. Interestingly, the small remaining effect seen with the combination appears to lack the antidepressant-like response observed with SNP alone, suggesting the involvement of alternative mechanisms. These pathways may include those unrelated to the serotonergic or NO-cGMP systems, such as the noradrenergic system, which can be influenced by the high polyphenol content of the aqueous extract of *P. granatum* [13].

To further assess whether *P. granatum* exerts antidepressant effects when NO synthesis is partially inhibited, we co-administered with NOS inhibitors 2.5 mg/kg for L-NAME or 40 mg/kg for 7-NI with *P. granatum* (1.0 mg/kg). L-NAME partially inhibited its antidepressant-like effect, particularly in the FST. This may reflect either L-NAME’s higher affinity for NO binding sites or a stress level that surpasses the adaptogenic capacity of *P. granatum* [28,39]. Remarkably, the tail suspension test is considered a moderate stressor, which may explain why effects were only detected in the FST [40].

Adaptogens are known to enhance nonspecific resistance to stress but may interfere pharmacokinetically with other compounds. In this context, competitive inhibition by L-NAME might displace *P. granatum*’s interaction with NOS [39]. Adaptogens also act preferentially under mild-to-moderate stress conditions, which may clarify why *P. granatum* loses effectiveness when combined with a prodepressant agent under high-stress conditions, as seen in the FST [28].

On the other hand, the co-administration of *P. granatum* and 7-NI (40 mg/kg) entirely abolished the antidepressant-like effects of *P. granatum* in both behavioral tests. Climbing and swimming behaviors decreased to levels observed with 7-NI alone. This suggests that *P. granatum* may primarily interact with nNOS, whose selective inhibition by 7-NI (unlike the broader inhibition by L-NAME) may directly interfere with *Punica granatum*’s mechanism of action. Higher doses of *P. granatum* might be needed to displace 7-NI from nNOS binding sites.

Although alternative mechanisms have been proposed for *P. granatum*’s antidepressant-like activity, these results reinforce the idea that its interaction with the NO-cGMP pathway involves a reduction in NO availability or downstream signaling. However, co-administration with NOS inhibitors at prodepressant doses failed to reverse their effects, possibly due to compensatory NO production by uninhibited isoforms, once again reflecting the pharmacokinetic complexity of adaptogenic compounds like *P. granatum*.

Physical exertion, such as swimming, increases oxidative stress through the generation of oxygen-free radicals [41], often detected via lipid peroxidation and altered glutathione metabolism. We therefore investigated whether *P. granatum* could protect against such oxidative damage. Swimming significantly decreased the GSH/GSSG ratio across brain regions and serum, indicating oxidative stress; however, pretreatment with *P. granatum* attenuated these effects. An increase in the GSH/GSSG ratio and a decrease in lipid peroxidation were observed in the cortex, hippocampus, brainstem, and serum, indicating a neuroprotective effect. These findings are consistent with previous studies on melatonin, a known antioxidant [42,43,44].

Phytochemical characterization confirmed that punicalagin and its metabolite ellagic acid are among the most abundant bioactive constituents of the *P. granatum* aqueous extract. It is well-known that punicalagin, as a hydrolysable tannin, possesses strong antioxidant capacity, as demonstrated in various assays [15,23,45] and compared to other nutraceuticals [46,47,48]. In this context, it is important to note that the extract used in the present study was obtained from the lyophilized juice of the whole fruit, rather than from the arils alone. The use of whole fruit ensures higher abundance and diversity of phenolic compounds, particularly punicalagins and ellagic acid derivatives [49,50]. This multimodal profile is consistent with previous reports, which show that both ellagic acid and punicalagin exert antidepressant and antioxidant-like effects in rodents [6,15]. Additionally, these ellagitannins reduce oxidative stress, which is known to contribute to neuronal cell death responsible for several neurodegenerative diseases [51,52], suggesting their involvement in the neuroprotective and antidepressant effects of *P. granatum*.

Beyond the nitrergic pathway, other neurotransmitter systems may also contribute to the antidepressant-like effects of *P. granatum*. For instance, noradrenergic and serotonergic contributions cannot be excluded, given the established role of these monoamines in stress regulation and depression-related behaviors. Additionally, the antioxidant properties of *P. granatum* may indirectly influence these neurotransmitter systems by modulating redox-sensitive signaling pathways. Therefore, the present findings should be interpreted within the broader context of possible multimodal mechanisms.

Limitations: A limitation of the present study is that we did not directly assess the activity of punicalagin and ellagic acid on the NO system. Consequently, it remains unknown whether either compound alone can reproduce the antidepressant-like effects observed with the complete *P. granatum* extract. In addition, our conclusions regarding nitrergic involvement are based on pharmacological evidence obtained with NO donors and NOS inhibitors and thus remain indirect. We did not measure specific markers of the NO pathway, such as nitrite/nitrate levels, NOS activity or expression, or cGMP signaling. Therefore, our interpretation should be regarded as inferential, and future studies incorporating biochemical and molecular assays will be essential to validate the proposed mechanisms.

Another limitation is that only male mice were studied. This choice was made to reduce variability associated with hormonal fluctuations of the estrous cycle; however, it restricts the generalizability of our findings. Sex differences are increasingly recognized in oxidative stress responses and depression-related behaviors, and future studies including female cohorts will be necessary to determine whether the antidepressant-like and antioxidant effects of *P. granatum* are sex-dependent in mice.

Finally, the experiments were conducted under an acute dosing paradigm. However, others report the chronic efficacy and safety profile of *P. granatum* in mice [53]. Further studies addressing long-term administration will be critical to strengthen the mechanistic and translational implications of our findings.

## 4. Materials and Methods

### 4.1. Animals

For the present research, adult male mice of three months of age from the Swiss Webster strain were provided from Cinvestav–Sede Sur vivarium. These mice were maintained at a room temperature of 22 ± 2 °C, under a 12-h light/12-h dark cycle, and a relative humidity of 50%. Food and water were available ad libitum, and they were housed in groups of five.

Tissue samples were immediately collected after decapitation of the animals and were preserved at a temperature of −70 °C. Animals not used for biochemical analyses were euthanized with CO_2_.

All experimental procedures with animals were carried out considering the guidelines established by the Institutional Ethics Committee for the Care and Use of Laboratory Animals (CICUAL Cinvestav; Protocol No. 0275-18) and the official Mexican Standard for the care and handling of animals (NOM-062-ZOO-1999).

### 4.2. Reagents and Treatment

All drugs were purchased from Merck Mexico and Sigma Aldrich (Toluca, Edo. de Mexico, México). Sodium nitroprusside (SNP) and *N*(G)-nitro-L-arginine methyl ester (L-NAME) were dissolved in 0.9% saline solution. 7-Nitroindazole (7-NI) was dissolved in a minimal volume of Tween 80 prior to administration. All compounds were administered intraperitoneally (i.p.) at a volume of 10 mL/kg body weight.

### 4.3. Preparation and Phytochemical Characterization of the P. granatum Extract

The aqueous extract of *P. granatum* was obtained from the lyophilized juice of the whole fruit. The use of the entire fruit, rather than arils alone, was chosen because the whole fruit provides a greater abundance and diversity of phenolic compounds, particularly ellagic acid derivatives and punicalagins, which appear in lower concentrations in aril-derived juice due to dilution with other ellagitannins and soluble sugars [49,50].

The presence of punicalagin isomers (α and β) and ellagic acid derivatives was confirmed by chromatographic analysis (Figure 1). The extract (6 mg/mL) was dissolved in HPLC-grade methanol, filtered through 0.22 μm GHP filters (Acrodisc 13, Waters, Milford, MA, USA), and analyzed on a Waters Acquity UPLC H-Class system equipped with a PDA detector and coupled to a mass spectrometer (UPLC-MS, Acquity Waters, Singapore). Data were processed with Empower software v.3 (Waters, Milford, MA, USA), following previously described protocols [15].

Phytochemical characterization revealed that ellagic acid derivatives (both free and glycosylated forms) and punicalagin isomers were the most abundant constituents, with quantified concentrations of 6.33 ± 0.04 mg/g for ellagic acid and 5.46 ± 0.04 mg/g for punicalagin (mg per g dry extract) [49]. These levels may vary depending on the pomegranate variety, cultivation region, and extraction method. Importantly, the present study used the complete extract rather than isolated compounds; therefore, the antidepressant-like effects observed are likely attributable to the synergistic action of these phenolics along with other minor constituents.

### 4.4. Behavioral Tests

(a)Tail suspension test (TST)

The TST is a behavioral assay used for screening potential antidepressant drugs [54] and measuring stress behavior in mice [22]. Mice were suspended from the tip of their tails at a height of 10 cm above the ground for five minutes, and the test was video recorded to evaluate the accumulated immobility time during the test. A decrease in immobility time indicates potential antidepressant effects [55].

(b)Forced swimming test (FST)

The Forced Swim Test (FST) is a well-established behavioral paradigm used to evaluate antidepressant-like effects in rodent models.

Animals were placed individually in acrylic cylinders with a diameter of 10 cm and a height of 25 cm, filled with 600 mL of water (23 ± 2 °C) [40]. The animals were allowed to swim for 6 min and were videotaped for later scoring by an experienced observer who was blind to the experimental manipulations. After the swimming test, the mouse was removed from the cylinders, dried with paper towels, and placed in heated cages. Thirty minutes later, it was returned to its home cage.

A time sampling technique was employed, in which, during the last 4 min of the FST, at the end of each 5-second interval, the behavior presented mainly by the mouse was scored [56]. Immobility was considered when the mice remained motionless in the water, making only the essential movements to keep their heads above the water. Swimming behavior was considered when the mouse made movements that allowed it to move around the cylinder. Climbing behavior was observed when the mouse made vigorous movements with all four limbs against the walls of the cylinder.

(c)Locomotor activity test

The locomotor test is used to rule out alterations in general activity that may interfere with performance on the behavioral tests. The apparatus consists of an acrylic box of 53 × 43 × 20 cm. The floor was divided into 12 equal squares. The mouse was placed in the corner of the box for 10 min. The number of crossed squares with all four legs was counted [57].

### 4.5. Lipid Peroxidation in the Mouse Brain

LPO in brain regions was assessed through the quantification of thiobarbituric acid reactive species (TBA-RS). Briefly, tissues were homogenized 1:10 with Krebs Buffer, which contains NaCl (19 mM), MgSO_4_ (1.2 mM), glucose (5 mM), NaH_2_PO_4_ (13 mM), Na_2_HPO_4_ (3 mM), KCl (5 mM), and CaCl_2_ (2 mM) in a final volume of 250 µL. Then, homogenates were mixed with 250 µL of thiobarbituric acid solution (TBA: 2.54% HCl, 0.375% TBA, and 15% trichloroacetic acid (TCA). For serum samples, 75 µL were mixed with 150 µL of TBA solution in a final volume of 225 µL. Samples were boiled in a water bath for 15 min, followed by centrifugation at 9800× *g* for 10 min. Optical density was measured at 532 nm using a Synergy H1 microplate reader. Results were expressed as µM of MDA (malondialdehyde) per milligram of protein or µM of MDA/µL in serum samples.

#### 4.5.1. Protein Determination

Briefly, 10 µL of samples were mixed with 1 mL of C solution (C= A + B solutions; A solution (2% of Na_2_CO_3_, 0.4% of NaOH, and 0.2% of sodium tartrate); and B solution (0.5% of Cu_2_(SO_4_)_3_) and incubated at room temperature for 10 min. After that, 100 µL of Folin solution (1:2) was added, and the samples were incubated at room temperature for 30 min. Absorbance was measured at 550 nm using a Synergy H1 microplate reader (BioTek Agilent, Santa Clara, CA, USA).

#### 4.5.2. Determination of GSH and GSSG in the Mouse Brain

Brain regions were homogenized in buffer A containing diethylenetriamine penta-acetic acid (DTPA, five mM), KCl (154 mM), and potassium phosphate buffer (0.1M, pH 6.8) at a 1:10 ratio. Then, an equal volume of buffer B, DTPA (20 mM), ascorbic acid (20 mM), trichloroacetic acid (TCA) (10%), and HCl (40 mM) was added to the homogenate. Serum samples were treated with metaphosphoric acid (5%, 1:4). The samples were then mixed and centrifuged for 20 min at 14,000× *g*. For GSH determination, 5 µL of supernatant was mixed with *O*-phthalaldehyde (OPA). For GSSG, 30 µL of supernatant was used. For GSSG, supernatants were mixed with *N*-ethylmaleimide (7.5 mM) to neutralize GSH; then, sodium hydrosulfite (100 mM) was added to reduce GSSG to GSH, and finally, OPA was added to facilitate isoindole formation. The fluorescence was measured at an excitation wavelength of 370 nm and an emission wavelength of 420 nm using a Synergy H1 microplate reader. The results were obtained using nmoles of GSH or GSSG per gram of tissue.

### 4.6. Experimental Design

#### 4.6.1. Experiments 1 and 2: Effects of *P. granatum*, SNP, L-NAME, and 7-NI Alone and in Combination on the TST and FST

Dose–response curves were constructed in independent groups of mice (*n* = 10 per group, Figure 13), and *P. granatum* was tested at doses of 0.25 mg/kg, 0.5 mg/kg, and 1.0 mg/kg. SNP as an NO donor, at doses of 1.0 mg/kg and 2.0 mg/kg; L-NAME as an NOS inhibitor at doses: 2.5 mg/kg, 5.0 mg/kg, 10 mg/kg, 20 mg/kg and 40 mg/kg; as well as 7-NI, a selective inhibitor of nNOS at doses: 20 mg/kg, 40 mg/kg, and 80 mg/kg were tested. The control group received a saline solution, and all compounds were administered intraperitoneally 30 min prior to the TST and FST to assess antidepressant-like effects. A locomotor activity test was conducted to assess motor function.

The latency time between administration and performance of behavioral tests was determined based on previous studies [11,31,33] (Figure 13).

For experiment 2, once the effective dose response for each drug was determined, the combined effect *P. granatum* (1.0 mg/kg) plus SNP (1.0 mg/kg), *P. granatum* (1.0 mg/kg) plus L-NAME (2.5 mg/kg), or *P. granatum* (1.0 mg/kg) plus 7-NI (40 mg/kg) were tested in independent group of mice (*n* = 10) subjected to TST, FST (the order of behavioral tests for experiments 1 and 2 was as follows: TST and FST) and locomotor activity (Figure 13).

#### 4.6.2. Experiment 3: Effect of *P. granatum* on GSH/GSSG and Lipoperoxidation in Hippocampus, Frontal Cortex, Amygdala, Brain Stem, and Serum

For experiment 3, the determination of oxidized glutathione and reduced glutathione was carried out in independent groups of mice (*n* = 5), in addition to the evaluation of LPO in the regions of the frontal cortex, hippocampus, amygdala, brain stem, as well as in the blood serum of mice treated with *P. granatum* (1 mg/kg), subjected, or not, to FST (Figure 13).

In all experiments, mice were randomly assigned to independent groups. Male mice were treated 30 min prior to the behavioral tests with saline, *P. granatum*, SNP, L-NAME, 7-NI, or their combinations. The latency between treatment and behavioral tests was determined based on previous studies [11,31].

### 4.7. Statistical Analysis

The G-power program (Apponic, version 3.1.9.7) was utilized to calculate the sample size, based on a previous study examining the effect of *P. granatum* on immobility behavior [11,31]. For this purpose, a reduction in immobility behavior by 60% with respect to the control group in the FST was considered as effective. Then, the G power program yielded a value size for a one-way ANOVA test of a total sample of 60 animals per dose response of *P granatum* with three doses and one control group. Following the recommendations to minimize the number of mice used as much as possible, we formed independent groups of 10 animals per group in all behavioral experiments. With this number per drug evaluation, we expected to reach a medium effect size of 0.57 with an α of 0.05 and power of 0.95. Following this approximation, the total sample size was 170 male mice for the dose–response curve, 120 for the combinations, and 20 for the LPO and GSH/GSSG assays.

Graphs were created using GraphPad Prism version 9.0.1, while statistical analyses were performed using SigmaPlot 12.3. Data were expressed as mean ± standard error, and the normality of the data was assessed using the Shapiro–Wilk test prior to conducting a one-way ANOVA to determine differences between treatments in behavioral experiments followed by a post hoc Dunnett’s or Tukey’s test, with a significance threshold set at *p* < 0.05. For lipid peroxidation and GSH/GSSG determination, a two-way ANOVA test was performed, followed by a post hoc Holm–Sidak test. All animals were included in the statistical analysis. In all cases, the size effect was calculated using eta2 (η^2^). A Pearson correlation analysis was performed between behaviors observed in FST and LPO and GSH/GSSG measures.

Dose–response data were analyzed using a logarithmic linear regression model to estimate the median effective dose (ED50). The slope of the regression was used to calculate the dose product.

## 5. Conclusions

This study demonstrates that *P. granatum* exhibits antidepressant-like effects associated with the attenuation of oxidative stress. Our behavioral data further suggest a possible involvement of nitric oxide (NO)-related mechanisms, as indicated by the modulatory effects observed with the co-administration of NO donors and NOS inhibitors. While these findings point toward an interaction with NO signaling, we acknowledge that direct measurements of nitrergic stress parameters under the combined treatments were not performed. Taken together, the results support the potential of *P. granatum* as a promising natural agent for the treatment of depression, particularly in conditions where oxidative imbalance and NO dysregulation are implicated.

## Figures and Tables

**Figure 1 ijms-26-10255-f001:**
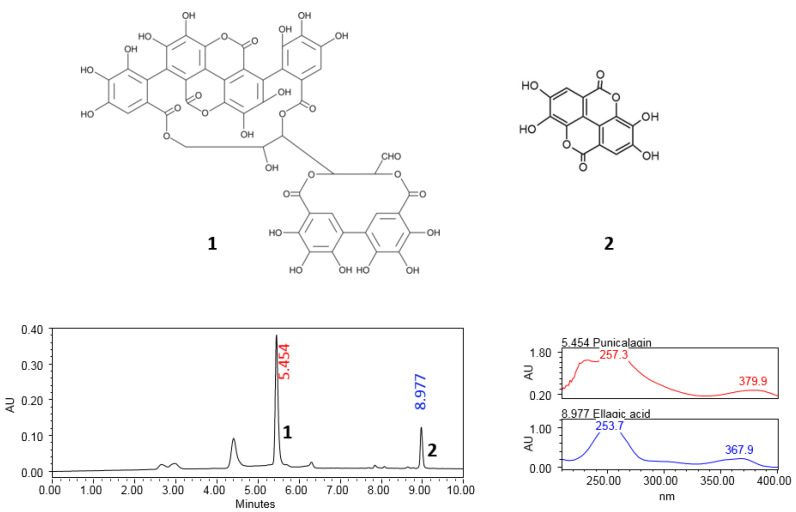
UPLC chromatographic profile of *P. granatum* aqueous extract showing the UV spectrum of two peaks related to the presence of punicalagin (**1**) and ellagic acid (**2**) compounds at the retention times of 5.454 min and 8.977 min, respectively.

**Figure 2 ijms-26-10255-f002:**
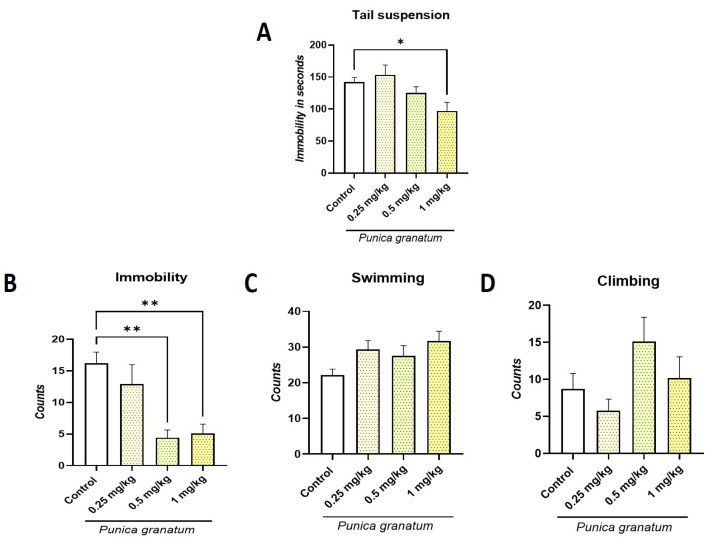
Determination of dose response for *P. granatum* (0.25, 0.5, and 1.0 mg/kg) in the tail suspension test (TST) and forced swim test (FST). Tail suspension (**A**), immobility behavior (**B**), swimming behavior (**C**), and climbing behavior (**D**). Data represent mean ± SEM; one-way ANOVA followed by Dunnett’s post hoc test, * *p* < 0.05 and ** *p* < 0.01, *n* = 10.

**Figure 3 ijms-26-10255-f003:**
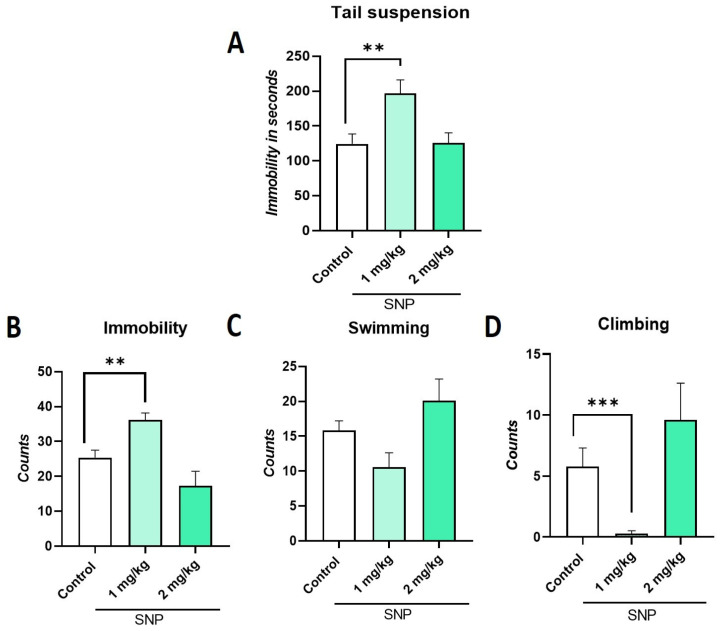
Determination of the dose response of the sodium nitroprusside SNP a nitric oxide donor (1.0 y 2.0 mg/kg), in the tail suspension test (**A**) and forced swim test behaviors such as immobility (**B**), swimming (**C**), and climbing (**D**). Data represent mean ± SEM; one-way ANOVA followed by Dunnett’s post hoc test, ** *p* < 0.01, *** *p* < 0.001, *n* = 10, SNP: sodium nitroprusside.

**Figure 4 ijms-26-10255-f004:**
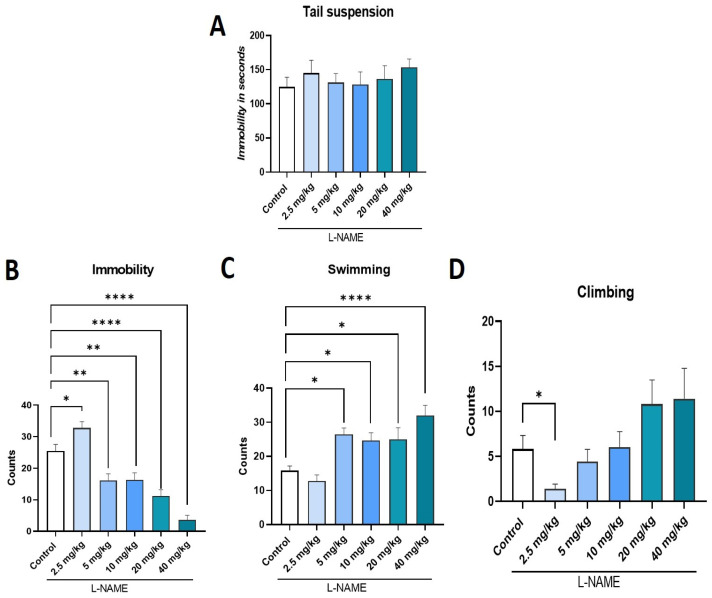
Dose response determination of the nitric oxide synthase inhibitor L-NAME (2.5, 5.0, 10, 20 y 40 mg/kg) in TST (**A**) and FST, immobility behavior (**B**), swimming behavior (**C**), and climbing behavior (**D**). Data represent mean ± SEM; one-way ANOVA followed by Dunnett’s post hoc test, * *p* < 0.05; ** *p* < 0.01, **** *p* < 0.001; *n* = 10, L-NAME, NG-Nitro-L-arginine methyl ester.

**Figure 5 ijms-26-10255-f005:**
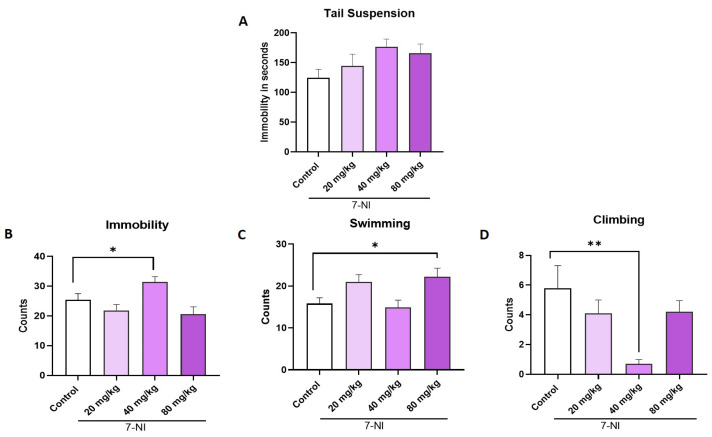
Identification of the dose response of the selective inhibitor of neuronal nitric oxide synthase 7-nitroindazole 7-NI (20, 40 y 80 mg/kg) in TST (**A**) and FST behaviors, immobility (**B**), swimming (**C**), and climbing (**D**). Data represent mean ± SEM; one-way ANOVA followed by Dunnett’s post hoc test, * *p* < 0.05 and ** *p* < 0.01, *n* = 10.

**Figure 6 ijms-26-10255-f006:**
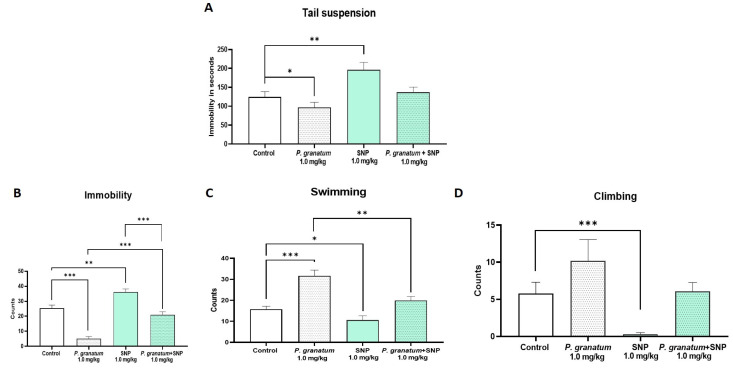
Effect of dose combination in independent groups of *P. granatum* (1.0 mg/kg) plus SNP (1.0 mg/kg), in the tail suspension test (**A**), and the forced swimming test, immobility (**B**), swimming (**C**), and climbing behavior (**D**). Data represent mean ± SEM, one-way ANOVA followed by Tukey’s post hoc test, * *p* < 0.05, ** *p* < 0.01, *** *p* < 0.001, *n* = 10, SNP: sodium nitroprusside.

**Figure 7 ijms-26-10255-f007:**
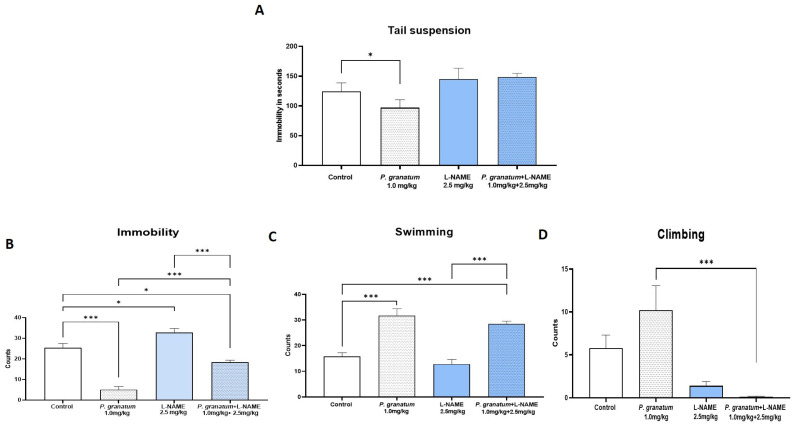
Effect of dose combination in independent groups of *P. granatum* (1.0 mg/kg) plus L-NAME (2.5 mg/kg), in the tail suspension test (**A**), and the forced swimming test, immobility (**B**), swimming (**C**), and results in the climbing behavior (**D**). Data represent mean ± SEM; one-way ANOVA followed by Tukey’s post hoc test. * *p* < 0.05, *** *p* < 0.001, *n* = 10, L-NAME: NG-Nitro-L-arginine methyl ester.

**Figure 8 ijms-26-10255-f008:**
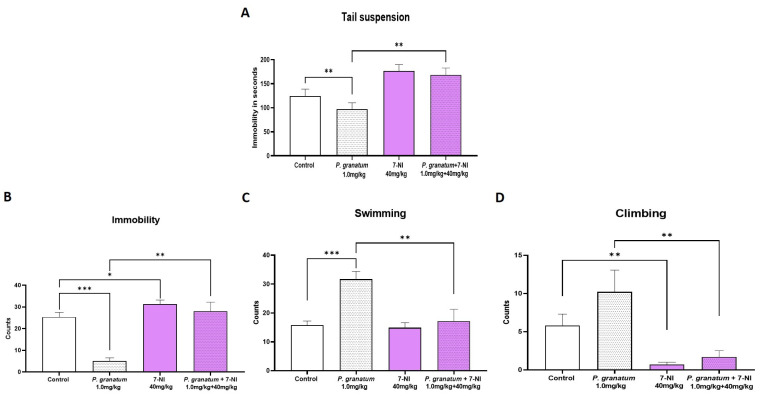
Effect of dose combination in independent groups of *P. granatum* (1.0 mg/kg) plus 7-NI (40 mg/kg), in the tail suspension test (**A**), and the forced swimming test, immobility (**B**), swimming (**C**), and climbing behavior (**D**). Data represent the mean ± SEM; one-way ANOVA followed by Tukey’s post hoc, * *p* < 0.05, ** *p* < 0.01, *** *p* < 0.001, *n* = 10, 7-NI: 7-nitroindazol.

**Figure 9 ijms-26-10255-f009:**
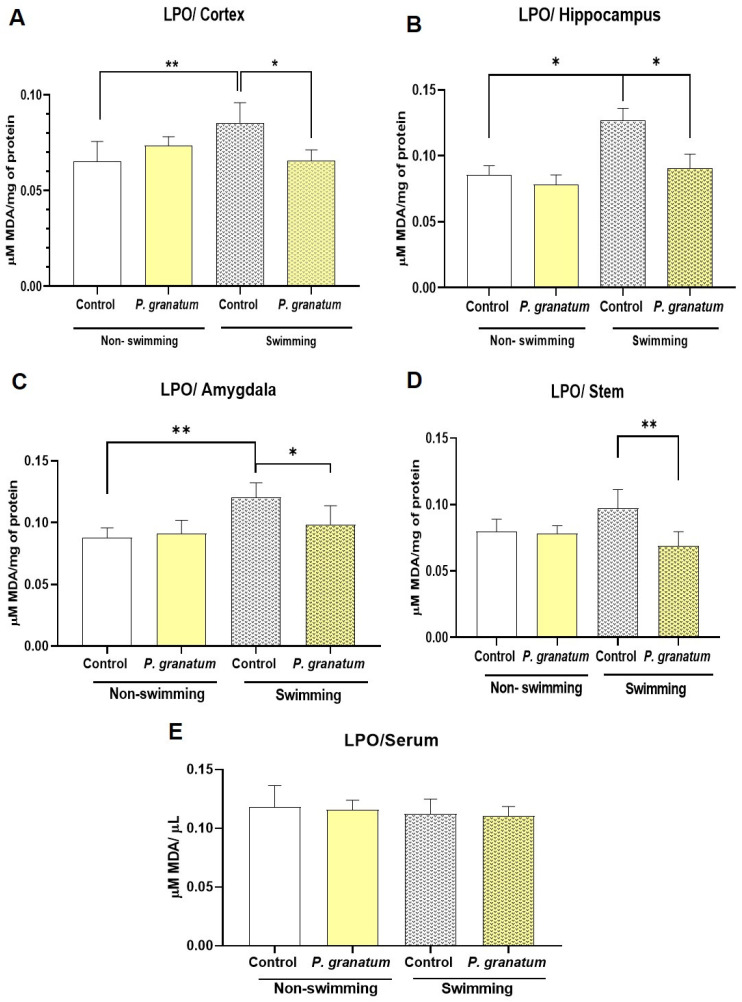
Determination of Lipid Peroxidation (LPO) in the Mice’s Brain. Graphs showing LPO determined in µM of malondialdehyde (MDA) per milligram of protein from mice treated with physiological saline solution (SSF) as a control group and *P. granatum* (1.0 mg/kg) with swimming and non-swimming in brain regions such as cortex (**A**), hippocampus (**B**), amygdala (**C**), stem (**D**), and LPO in serum (**E**). Data represent mean ± SEM; Two-way ANOVA followed by Holm–Sidak post hoc test, * *p* < 0.05, ** *p* < 0.001 versus control group, *n* = 5.

**Figure 10 ijms-26-10255-f010:**
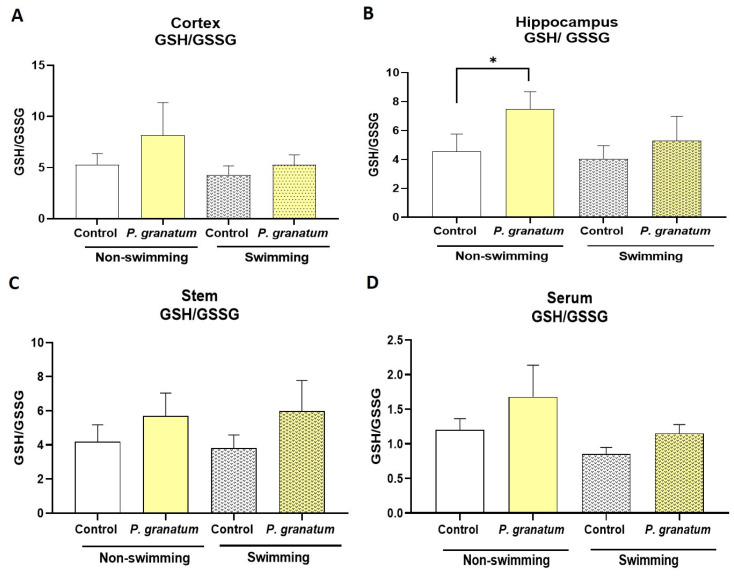
Measurement of the oxidized/reduced glutathione ratio in mice brains. Treated with vehicle (SSF) as a control group and with *P. granatum* (1.0 mg/kg) with and without swimming for both treatment groups in regions of cortex (**A**), hippocampus (**B**), stem (**C**), and serum (**D**). Data represent mean ± SEM; Two-way ANOVA followed by Holm–Sidak post hoc test, * *p* ≤ 0.01 versus control group, *n* = 5.

**Figure 11 ijms-26-10255-f011:**
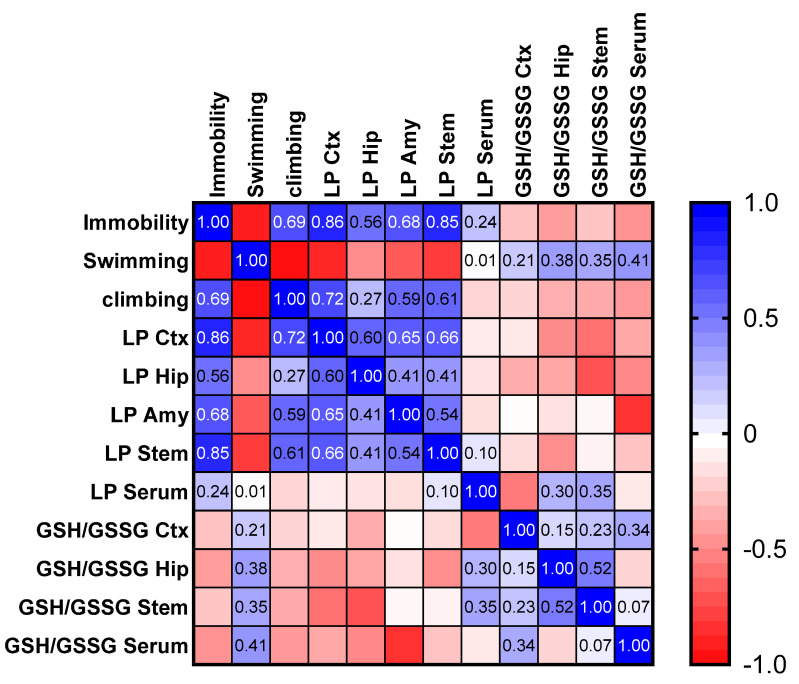
Pearson correlation between FST behavior (immobility, swimming and climbing) and lipid peroxidation (LP) and oxidative stress index (GSH/GSSG). Ctx= frontal cortex; Hip = hippocampus; Amy = amygdala; Stem = brain stem.

**Figure 12 ijms-26-10255-f012:**
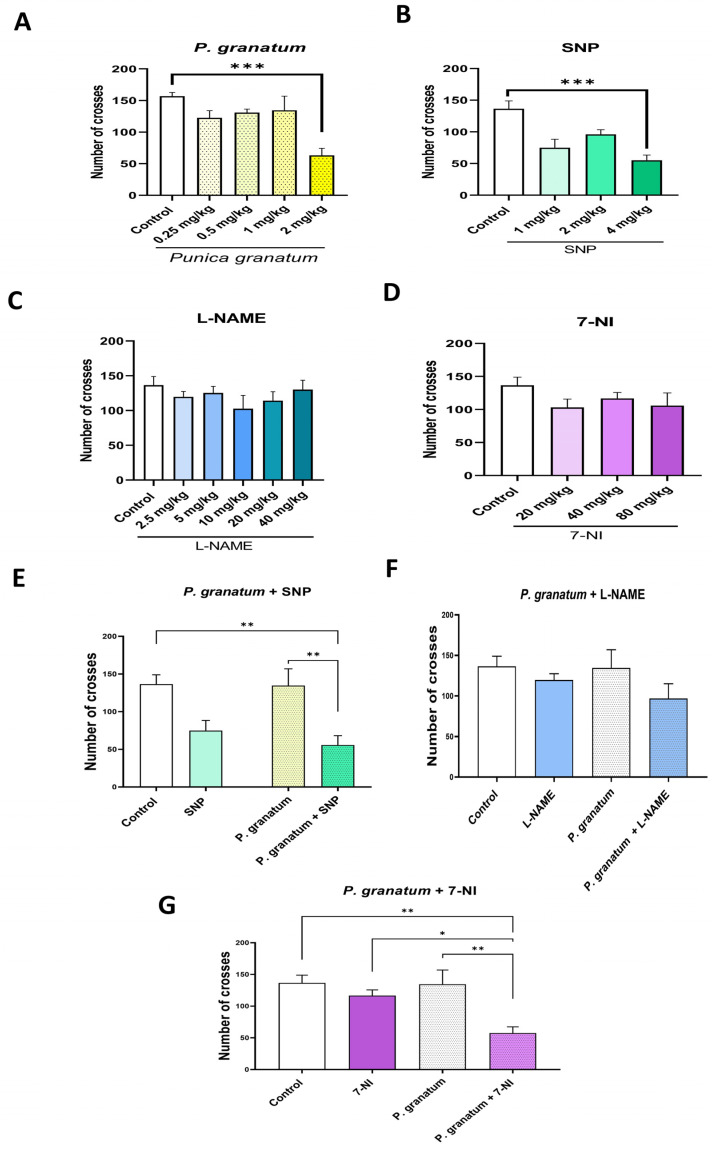
Determination of the dose–response for the different study treatments, (**A**) *P. granatum* (0.25, 0.5, 1.0 y 2.0 mg/kg), (**B**) SNP (1, 2 y 4 mg/kg), (**C**) L-NAME (2.5, 5.0, 10, 20 y 40 mg/kg), (**D**) 7-NI (20, 40 y 80 mg/kg), (**E**) PG (1.0 mg/kg) + SNP (1.0 mg/kg), (**F**) PG (1.0 mg/kg) + L-NAME (2.5 mg/kg), and (**G**) PG (1.0 mg/kg) + 7NI (40 mg/kg) in the locomotor activity test. Data represent mean ± SEM; one-way ANOVA, *n* = 10. Tukey test: * *p* ≤ 0.05, ** *p* < 0.01 and *** *p* < 0.001.

**Figure 13 ijms-26-10255-f013:**
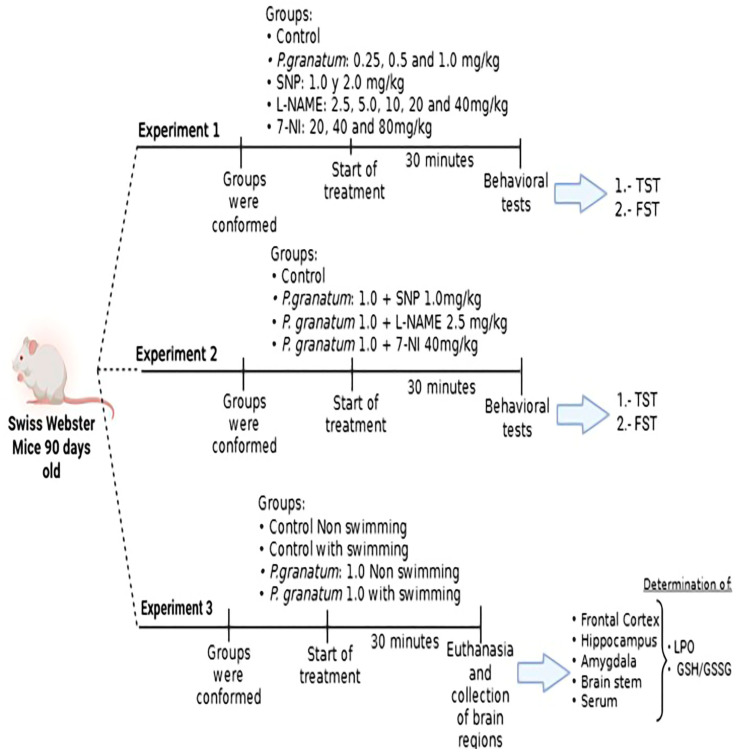
Schematic representation of the experimental design. SNP = sodium nitroprusside; L-NAME = NG-nitro-L-arginine methyl ester; 7-NI= 7-nitroindazol; GSH/GSSG = oxidized/reduced glutathione ratio; LPO = lipoperoxidation.

**Table 1 ijms-26-10255-t001:** One-way ANOVA values for locomotor activity.

Test	Variable	Factor Treatment	One-Way ANOVA Values	*p*-Value
Locomotor activity	Number of crosses	Punica granatum	F(4, 45) = 7.485	0.0001
		SNP	F(3, 36) = 10.73	<0.0001
		L-NAME	F(5, 54) = 0.8505	ns
		7-NI	F(3, 35) = 1.207	ns
		Punica granatum + SNP	F(5, 54) = 7.211	<0.0001
		Punica granatum + L-NAME	F(3, 36) = 1.288	ns
		Punica granatum + 7-NI	F(3, 36) = 8.726	0.0002

## Data Availability

The raw data supporting the conclusions of this article will be made available by the authors upon reasonable request.

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
