# Peer review of "Role of Nitric Oxide in the Antidepressant Effect of an Aqueous Extract of Punica granatum L.: Effects on GSH/GSSG Ratio and Lipoperoxidation in Adult Male Swiss Webster Mice"

_ijms, 2025, doi:10.3390/ijms262110255_

Round 1
Reviewer 1 Report
Comments and Suggestions for Authors
In the paper of Cervantes-Anaya et al. authors studied the antidepressant effects of water extract of Punica granatum in mice. They found that Punica granatum showed the antidepressant-like effects in male Swiss Webster mice through a mechanism involving attenuation of oxidative and nitrergic stress.
Despite the extensive experimental work that was carried out, I have serious complaints about the article.
The main thing is that the authors studied the aqueous extract of pomegranate, which was obtained from the lyophilized juice of whole fruits. Obviously, the result of extracting lyophilized juice and whole fruit should be different, but nothing is said about this. However, this is not my main question.
The authors studied the effect of the aqueous extract at a dose of 0.25-2 mg / kg, but it is not clear what is there. One or two milligrams of what? The authors found that the extract contains two active ingredients, by which of them was the effect assessed? Or by the sum of both? How was the extract standardized? Without a clear understanding of what is in the aqueous extract and how it is standardized by the concentration of the active substance, it is difficult to evaluate the study.
There are other complaints about the article.
- None of the tests used by the authors (FST and TST) are described in the methods. It is not clear what was done and how the parameters indicated in the figures were measured.
- The methods indicate that the extract was used in doses of 0.25-2 mg/kg, but the figures do not show results for a dose of 2 mg/kg
- Aqueous solutions of the studied compounds were administered intraperitoneally in a volume of 2 ml/kg, it seems to me that this is a very small volume (0.04 ml per 20-gram-mouse), it is very difficult to dose
- The authors wrote that they evaluated the Dose-response curves, usually this should end with calculating ED50? For some reason, the authors did not do this.
- How was the level of reduced glutation (GSH) and oxidized glutathione (GSSG) quantitatively assessed?
- How was lipid peroxidation quantitatively assessed?
- The figures with the combined use of substances repeat the results presented in the previous figures.
Author Response
1 - The authors studied the aqueous extract of pomegranate, which was obtained from the lyophilized juice of whole fruits. Obviously, the result of extracting lyophilized juice and whole fruit should be different, but nothing is said about this.
R- The decision to study the extract obtained from the lyophilized juice of whole fruits was based on its phytochemical richness. The importance of using the whole fruit rather than only freeze-dried aril juice lies in the higher abundance and diversity of phenolic compounds, particularly ellagic acid derivatives and punicalagins (González-Trujano et al., 2015). While both compounds can be found in juice and whole fruit, their concentration is lower when only the arils are used, since dilution occurs among other ellagitannins and water-soluble sugars. By contrast, the whole fruit extract preserves higher levels of punicalagins and ellagic acid derivatives, which are considered authenticity markers of pomegranate extracts (Mena et al., 2011). We have clarified this point in the revised manuscript in the Methods and Discussion sections.
2- The authors studied the effect of the aqueous extract at a dose of 0.25-2 mg / kg, but it is not clear what is there. One or two milligrams of what? The authors found that the extract contains two active ingredients, by which of them was the effect assessed? Or by the sum of both? How was the extract standardized? Without a clear understanding of what is in the aqueous extract and how it is standardized by the concentration of the active substance, it is difficult to evaluate the study.
R- We thank the reviewer for bringing this important issue to our attention. In our extract, the main phenolic compounds identified were ellagic acid derivatives (both the glucoside and the free form) and the α- and β-isomers of punicalagin. The total concentrations were previously quantified as mg of compound per g of dry extract: 6.33 ± 0.04 mg/g for ellagic acid and 5.46 ± 0.04 mg/g for punicalagin isomers, respectively (González-Trujano et al., 2015). We acknowledge that these values may vary depending on the fruit variety, geographic origin, and extraction efficiency. In the present study, the extract was administered as a complete preparation, not as isolated compounds; therefore, the antidepressant-like effects cannot be attributed exclusively to ellagic acid or punicalagin, but rather to their combined action, along with other minor constituents. This multimodal profile is supported by prior evidence showing that both ellagic acid and punicalagin display antidepressant- and antioxidant-like effects in rodents (Girish et al., 2012; Cervantes-Anaya et al., 2022). We have clarified this point in the revised manuscript in the Methods and Discussion sections.
3 - None of the tests used by the authors (FST and TST) are described in the methods. It is not clear what was done and how the parameters indicated in the figures were measured.
R-We apologize for the oversight in our previous manuscript. The revised version now includes the missing information in the Methods section.
4 - The methods indicate that the extract was used in doses of 0.25-2 mg/kg, but the figures do not show results for a dose of 2 mg/kg.
R - The activity test was conducted to eliminate any nonspecific effects of the drugs under investigation. During this test, it was observed that a dose of 2 mg/kg of the extract impacted general activity. Consequently, this dose was not used in subsequent behavioral tests. This information is now presented in results section.
5 - Aqueous solutions of the studied compounds were administered intraperitoneally in a volume of 2 ml/kg, it seems to me that this is a very small volume (0.04 ml per 20-gram-mouse), it is very difficult to dose.
R - We apologize for the error; all administrations were conducted at a volume of 10 ml/kg. This has already been corrected in the methods section.
6 - The authors wrote that they evaluated the Dose-response curves, usually this should end with calculating ED50? For some reason, the authors did not do this.
R – We appreciate the reviewer’s comment. We have now included the information regarding the calculation of the ED50 for PunicagranatumL in both the Methods and Results sections. However, we did not report ED50 values for the other pharmacological agents tested, as the primary objective of our study was to investigate the potential role of nitric oxide (NO) pathways in the antidepressant-like effects of P. granatum.
In this context, we defined the effective dose of P. granatum as the dose that produced a significant reduction in immobility (in both the FST and TST) and an increase in swimming behavior (in the FST). The dose–response analyses of SNP (NO donor), L-NAME (non-selective NOS inhibitor), and 7-NI (selective nNOS inhibitor) were performed only to establish their modulatory actions in the FST and TST before exploring their interaction with P. granatum. Since these compounds exhibited biphasic effects, which would require a separate in-depth study, we used only the effective doses that produced a pro-depressive effect in the FST to evaluate whether P. granatum could reverse it.
To clarify this rationale, we have expanded the explanation in the Introduction section:“In addition, to better characterize the pharmacological profile of P. granatum, we determined the median effective dose (ED50) for its antidepressant-like effects in the FST.”
Furthermore, the Statistical Analysis section now specifies:“Dose–response data were analyzed using a logarithmic linear regression model to estimate the effective dose 50 (ED50). The slope of the regression was used to calculate the dose producing 50% of the maximal effect (ED50) for P. granatum in the FST.”
Correspondingly, the Results section reports:“The dose–response curve for P. granatum revealed a significant antidepressant-like effect, with an estimated ED50 of 0.45 mg/kg in the FST.”
7 - How was the level of reduced glutation (GSH) and oxidized glutathione (GSSG) quantitatively assessed?
R – We apologize for the oversight in our previous manuscript. The revised version now includes the methods for quantifying GSH and GSSG, which are described in the “Methods” section.
8 - How was lipid peroxidation quantitatively assessed?
R – Additionally, the revised manuscript now includes methods for quantifying lipid peroxidation in the “Methods” section.
9 - The figures with the combined use of substances repeat the results presented in the previous figures.
R – All experimental groups were independent; therefore, none of the experiments or their respective figures repeated results.
Reviewer 2 Report
Comments and Suggestions for Authors
In the present study, the potential antidepressant effect of an aqueous Pomegranate extract and its potential mechanism of action were investigated. Therefore, the behavior of mice in two predictive tests—the TST and the FST—after administration of the extract alone and in combination with NO pathway modulators - was assessed. Biochemical studies were also carried out to determine the level of oxidative stress markers in various brain structures of mice after administration of Pomegranate extract. The results may be relevant to studies on new antidepressant agent and its molecular mechanism. However, some issues raised my concerns, which I have listed below:
- The authors mention locomotor activity in line 400, but their results are never shown. This is a crucial parameter, as the influence of the compounds tested on motor functions determines the reliability of the presented results.
- The graphs showing the results of the tail suspension test should include the name of the measured parameter, i.e. immobility, and not just the units of time.
- On what basis were the doses of the tested extract selected for the presented studies?
- The diagram in figure 11 does not accurately represent the sequence of events during the experiments i.e. a) at what time intervals were the extract and NO modulators administered?; b) in what order were the tests carried out - TST and FST.
- In the conclusion, the Authors write about the contribution of nitrergic stress (line 422) to the effects of the tested extract. This is a far-fetched conclusion, as the presented study only presents the results of behavioral studies following the combined administration of the extract and NO modulators. Stress parameters in the mouse brain were not examined in the above experimental design. Stress parameters were only examined after the administration of the extract (compared to Fig. 9 and 10).
Corrections:
- I suggest supplementing the information about the form of the tested plant, i.e. please emphasize that it was an aqueous extract in the abstract and in the title.
- Line 267: this statement is an oversimplification.
- 11 “amygdala” not amigdala
Author Response
1 - The authors mention locomotor activity in line 400, but their results are never shown. This is a crucial parameter, as the influence of the compounds tested on motor functions determines the reliability of the presented results.
R - We appreciate the reviewer’s comment and apologize for omitting this information. In the new version of the manuscript, we have presented the methodology and results of the activity test.
2 - The graphs showing the results of the tail suspension test should include the name of the measured parameter, i.e. immobility, and not just the units of time.
R- Figures were corrected as suggested by the reviewer
3 - On what basis were the doses of the tested extract selected for the presented studies?
R – The doses were selected based on previous studies [9; 22; 31]. This information is now found in the experimental design section of the article.
4 - The diagram in figure 11 does not accurately represent the sequence of events during the experiments i.e. a) at what time intervals were the extract and NO modulators administered? b) in what order were the tests carried out - TST and FST.
R – Figure 13 (formerly 11) was corrected according to the reviewer's comment.
5 - In the conclusion, the Authors write about the contribution of nitrergic stress (line 422) to the effects of the tested extract. This is a far-fetched conclusion, as the presented study only presents the results of behavioral studies following the combined administration of the extract and NO modulators. Stress parameters in the mouse brain were not examined in the above experimental design. Stress parameters were only examined after the administration of the extract (compared to Fig. 9 and 10).
R –We agree with the reviewer, so the conclusion was changed as follows:
This study demonstrates that P. granatum exhibits antidepressant-like effects associated with the attenuation of oxidative stress. Our behavioral data further suggest a possible involvement of nitric oxide (NO)-related mechanisms, as indicated by the modulatory effects observed with the co-administration of NO donors and NOS inhibitors. While these findings point toward an interaction with NO signaling, we acknowledge that direct measurements of nitrergic stress parameters under the combined treatments were not performed. Taken together, the results support the potential of P. granatum as a promising natural agent for the treatment of depression, particularly in conditions where oxidative imbalance and NO dysregulation are implicated.
Corrections:
6 - I suggest supplementing the information about the form of the tested plant, i.e. please emphasize that it was an aqueous extract in the abstract and in the title.
R – The changes were made as indicated by the reviewer.
7 -Line 267: this statement is an oversimplification.
R -We replace this sentence by other to be more specific... A more complex mechanism was observed with the selective nNOS inhibitor 7-NI. Specifically, a dose of 40 mg/kg increased immobility, while doses of 20 and 80 mg/kg reduced it, suggesting a dose-dependent modulation of the monoaminergic pathways
Reviewer 3 Report
Comments and Suggestions for Authors
Dear Authors,
Thank you for the opportunity to review your manuscript investigating the antidepressant-like effects of Punica granatum (PG) and the putative involvement of nitric oxide (NO) and redox signaling in adult male Swiss Webster mice. The topic is timely and relevant to IJMS readers at the interface of neuropsychopharmacology, redox biology, and natural products. I appreciate the multi-layered design combining behavior (TST/FST) with biochemical readouts (lipid peroxidation, GSH/GSSG) and pharmacological probing of nitrergic pathways (SNP, L-NAME, 7-NI). The qualitative chromatographic identification of punicalagin and ellagic acid also adds plausibility regarding antioxidant capacity. At the same time, several aspects require substantial strengthening before the work can be considered for publication. Most importantly, your mechanistic conclusions about “nitrergic” mediation remain indirect: the manuscript lacks direct measurements of NO pathway indices (e.g., nitrite/nitrate, NOS activity or expression, cGMP), which currently makes the NO conclusions inferential. Likewise, the extract characterization is qualitative; quantitative standardization of punicalagin/ellagic acid (mg/g, % w/w with validation) is essential for reproducibility and translational relevance. Statistical reporting should include effect sizes and 95% CIs, control for multiplicity across numerous regions and endpoints, and provide exact p-values. Please also resolve the inconsistency between the G*Power calculation (stated total N=56/experiment) and the implemented group sizes, and justify the final sample sizes with power estimates tied to specific primary outcomes. Because TST/FST can be confounded by altered locomotion, the promised locomotor data must be fully reported (plots, statistics) and explicitly integrated into interpreting “swimming/climbing” changes. I further note that only male mice were studied; please discuss the sex limitation and, if feasible, consider adding a female cohort or clarify why this was impossible. The Abstract and Conclusions should be moderated to reflect the current evidential level (behavior + redox, no direct NO measures). Please perform thorough language editing to correct typographical and nomenclature inconsistencies. In Methods, please detail randomization, blinding procedures beyond scorer masking, water temperature, and illumination for FST, acclimatization, and humane endpoints. Finally, I encourage you to expand the Discussion with a balanced consideration of alternative pathways (e.g., noradrenergic contributions) and a dedicated limitations paragraph (indirect NO readouts, male-only cohort, acute dosing). Please address the points above in your rebuttal and highlight all textual changes in the revised manuscript.
Yours sincerely,
The reviewer.
Author Response
1- Most importantly, your mechanistic conclusions about “nitrergic” mediation remain indirect: the manuscript lacks direct measurements of NO pathway indices (e.g., nitrite/nitrate, NOS activity or expression, cGMP), which currently makes the NO conclusions inferential.
R – We appreciate the reviewer’s comment. We fully acknowledge that our conclusions regarding the involvement of the nitrergic pathway are based on pharmacological manipulations rather than direct biochemical or molecular measurements. While the use of NOS inhibitors, a NO donor, and their behavioral outcomes provides valuable inferential evidence, we agree that this does not substitute for direct quantification of NO pathway markers such as nitrite/nitrate levels, NOS activity or expression, or cGMP signaling.
To address this point, we have revised the Discussion to explicitly acknowledge this limitation. We also emphasize in the revised text that our findings should be interpreted as suggestive, but not definitive, evidence of nitrergic involvement. Finally, we propose that future studies incorporate direct biochemical and molecular assays to validate the mechanistic hypotheses raised in this work. (see limitations section)
2 - Likewise, the extract characterization is qualitative; quantitative standardization of punicalagin/ellagic acid (mg/g, % w/w with validation) is essential for reproducibility and translational relevance.
R - We appreciate the comment and have included some lines on materials and methods, results and/or discussion to reinforce the information provided.
We consider that punicalagin and ellagic acid are among the most important bioactive components responsible for the effects observed in the extract. For example, we know that at least 5-6 ug are present in a 1 mg dose of extract, and in previous studies we have found evidence that a dose of 10 ug/kg of ellagic acid or punicalagin for 14 days produced a significant response similar to that obtained with 1 mg/kg of the extract (Cervantes-Anaya et al., 2022).
3 - Statistical reporting should include effect sizes and 95% CIs, control for multiplicity across numerous regions and endpoints, and provide exact p-values.
We appreciate the reviewer’s comments and the opportunity to incorporate this information. We calculate η², which is the appropriate measure for determining effect size following a factorial analysis. As noted in the results section, we provide F values accompanied by p values and η² values. In instances where the data are statistically significant (as indicated by p-values), we also observe high η² values. According to the guidelines for effect size using η², values of 0.01 indicate a small effect, 0.06 a medium effect, and 0.14 a large effect. In our analysis, we observed effect sizes ranging from 0.40 to 0.80 in some cases.
4 - Please also resolve the inconsistency between the G*Power calculation (stated total N=56/experiment) and the implemented group sizes, and justify the final sample sizes with power estimates tied to specific primary outcomes.
R – Thanks for the opportunity to clarify this point. In our estimation of sample size we use the reduction of immobility behavior induced by P. granatum in the FST as the main outcome (Valdés-Sustaita et al 2017). For this purpose, a reduction of Immobility behavior by 60% with respect to the control group in the FST was considered as effective. Then, the G power program yielded a value size for a one way ANOVA test of a total sample of 60 animals per dose response of P granatum with three doses and one control group. Following the recommendations to minimize the number of mice used as much as possible, we formed independent groups of 10 animals per group in all behavioral experiments. With this number per drug evaluation, we expected to reach a medium effect size of 0.57 with an α of 0.05 and power of 0.95.
Following this approximation, the total sample size was 170 male mice for the dose-response curve, 120 for the combinations, and 20 for the LPO and GSH/GSSG assays.
This information is stated in the methods section.
5 - Because TST/FST can be confounded by altered locomotion, the promised locomotor data must be fully reported (plots, statistics) and explicitly integrated into interpreting “swimming/climbing” changes.
R - Complete information on locomotor activity has been added to this new version of the manuscript and is discussed in relation to the results of behavioral testing.
6 - I further note that only male mice were studied; please discuss the sex limitation and, if feasible, consider adding a female cohort or clarify why this was impossible.
R - We thank the reviewer for this observation. Indeed, only male Swiss Webster mice were included in the present experiments. This choice was made to minimize the influence of estrous cycle–related hormonal fluctuations, which can introduce variability in behavioral and oxidative stress parameters.
We agree that this represents a limitation, as the findings cannot be directly generalized to female animals. We have revised the Discussion to acknowledge the sex bias in the limitations section explicitly and to emphasize that future work should incorporate both sexes to evaluate potential sex-dependent differences in the antidepressant-like effects and oxidative stress responses to P. granatum.
7 -The Abstract and Conclusions should be moderated to reflect the current evidential level (behavior + redox, no direct NO measures).
R – The abstract and conclusions were rewritten following the reviewers' comments.
The revised Abstract has the following structured format (Background → Objective → Methods → Results → Conclusion); we try to improve clarity and conciseness. The new version highlights the rationale, experimental approach, main behavioral and biochemical findings, and a moderated conclusion. We also softened the claims regarding nitrergic stress, now stating that the involvement of NO pathways is suggested but remains indirect due to the lack of direct biochemical measurements.”
8 - Please perform thorough language editing to correct typographical and nomenclature inconsistencies.
R –The language was revised as suggested
9 -In Methods, please detail randomization, blinding procedures beyond scorer masking, water temperature, and illumination for FST, acclimatization, and humane endpoints.
R –We apologize for the omissions in the Methods section. The current version of the manuscript describes the methodology requested by all the reviewers.
10 - Finally, I encourage you to expand the Discussion with a balanced consideration of alternative pathways (e.g., noradrenergic contributions) and a dedicated limitations paragraph (indirect NO readouts, male-only cohort, acute dosing). Please address the points above in your rebuttal and highlight all textual changes in the revised manuscript.
R - We thank the reviewer for this valuable suggestion. In response, we have expanded the Discussion to provide a more balanced perspective by considering alternative pathways that could contribute to the antidepressant-like effects of P. granatum, including noradrenergic and serotonergic modulation, in addition to its known antioxidant properties.
Furthermore, we have now added a dedicated Limitations section at the end of the Discussion. This section explicitly acknowledges that: (i) our conclusions regarding nitrergic involvement are indirect and based on pharmacological manipulations without direct biochemical readouts; (ii) only male mice were studied, which restricts generalizability to females; and (iii) the present study was limited to acute dosing, so the long-term efficacy and safety of P. granatum remain to be established.
Reviewer 4 Report
Comments and Suggestions for Authors
Abstract
The abstract could be more direct and concise, highlighting the background (brief contextualisation on depression, the role of oxidative stress and NO), study objective, methods, main behavioural and biochemical findings, and conclusion. It would be useful to organise it into short sentences with clear transitions (Objective → Methods → Results → Conclusion).
Introduction
The text mixes epidemiology of depression, biological mechanisms and the effects of pomegranate in a diffuse way. It is suggested to reorganise it into three clear blocks:
-
Clinical relevance: prevalence of depression, limitations of current antidepressants.
-
Mechanistic basis: oxidative stress and the role of NO in depression (only the essential points).
-
Previous evidence: antidepressant effects of pomegranate and the gap — lack of studies on the involvement of NO.
Methods
The methods section should appear immediately after the introduction, before the results.
Results
Absence of effect sizes; only p-values are presented, without effect sizes or confidence intervals. Partial η² or Cohen’s d should be added, at least for the main results.
Behavioural and biochemical findings are presented in isolated blocks; there is no direct cross-link between them. Exploratory correlation analyses should be included (e.g., GSH/GSSG vs immobility in the FST).
Some figures are overloaded (too many subgraphs per figure). It is suggested to split them into supplementary panels.
Discussion
Many parts repeat the results without adding interpretation. The discussion should be synthesised, focusing on the novel findings and their contribution to the field.
Many references are old (>5 years). Recent reviews (2020–2024) on oxidative stress, NO and phytotherapeutics in depression should be included.
The discussion does not address relevance to humans (e.g., consumption of pomegranate juice, equivalent doses, limitations of extrapolation). A paragraph should be added on clinical implications and the need for clinical trials.
Limitations are underdeveloped, only mentioning that punicalagin/ellagic acid were not tested separately. Additional limitations should be included: use of males only (no consideration of sex differences); acute model (does not assess chronic administration); small biochemical samples (n=5); absence of behavioural markers of anhedonia (only FST/TST).
Conclusion
The conclusion states that pomegranate “represents a promising agent for the treatment of depression” without any caveats. The wording should be moderated, acknowledging that the results are pre-clinical in a murine model.
There is no direct link back to the hypotheses. The conclusion could explicitly return to the initial hypotheses, confirming or refuting each.
There is little guidance for future research. It only suggests studying isolated compounds. Recommendations should also include chronic studies, inclusion of females, analysis of translational doses, and clinical trials.
Author Response
1 - The abstract could be more direct and concise, highlighting the background (brief contextualisation on depression, the role of oxidative stress and NO), study objective, methods, main behavioural and biochemical findings, and conclusion. It would be useful to organise it into short sentences with clear transitions (Objective → Methods → Results → Conclusion).
R – As mentioned earlier, the abstract was revised in response to the reviewers' comments.
The revised Abstract has the following structured format (Background → Objective → Methods → Results → Conclusion); we try to improve clarity and conciseness. The new version highlights the rationale, experimental approach, main behavioral and biochemical findings, and a moderated conclusion. We also softened the claims regarding nitrergic stress, now stating that the involvement of NO pathways is suggested but remains indirect due to the lack of direct biochemical measurements.”
2 -Introduction: The text mixes epidemiology of depression, biological mechanisms, and the effects of pomegranate in a diffuse way. It is suggested to reorganize it into three clear blocks:
- Clinical relevance: prevalence of depression, limitations of current antidepressants.
- Mechanistic basis: oxidative stress and the role of NO in depression (only the essential points).
- Previous evidence: antidepressant effects of pomegranate and the gap — lack of studies on the involvement of NO.
R - We thank the reviewer for this valuable suggestion. In the revised version, the Introduction has been reorganized into three clearer sections as recommended: 1st- Clinical relevance, outlining the prevalence of depression and the limitations of current antidepressants. 2nd- Mechanistic basis, briefly summarizing the role of oxidative stress and nitric oxide (NO) in depression.3rd - Previous evidence and knowledge gap, highlighting the antidepressant-like effects of P. granatum, its antioxidant and NO-inhibitory properties, and emphasizing the lack of studies addressing the role of NO in its in vivo effects.
3 – Methods: The methods section should appear immediately after the introduction, before the results.
R - The organization of the manuscript is based on the journal's instructions.
4 – Results: Absence of effect sizes; only p-values are presented, without effect sizes or confidence intervals. Partial η² or Cohen’s d should be added, at least for the main results.
R - We appreciate the reviewer’s suggestion. In this version, we incorporated the F values, followed by the p and η² values. This information can be observed in the results section description.
5 - Behavioural and biochemical findings are presented in isolated blocks; there is no direct cross-link between them. Exploratory correlation analyses should be included (e.g., GSH/GSSG vs immobility in the FST).
R - Thank you for the opportunity to extend our analysis. In this version, we include a Pearson correlation between the variables measured in the Forced Swim Test (FST), lipid peroxidation (LPO), and the oxidative stress index (GSH/GSSG). As shown in figure 11, immobility behavior exhibits a positive correlation with LPO in the frontal cortex, hippocampus, and amygdala. In contrast, swimming behavior displays a slight positive correlation with GSH/GSSG. Climbing behavior, however, shows no association with either LPO or oxidative stress. The present data reinforce the suggestion that P. granatum mitigates the effects of lipid peroxidation induced by the FST and reduces oxidative stress.
6-Some figures are overloaded (too many subgraphs per figure). It is suggested to split them into supplementary panels.
R - We thank the reviewer for the suggestion. While we understand the concern, we feel that the current figure layout helps present the data in a more integrated and comparative way. Separating the panels might reduce clarity and context. We believe this format best serves the reader’s understanding of the data.
7 – Discussion: Many parts repeat the results without adding interpretation. The discussion should be synthesized, focusing on the novel findings and their contribution to the field.
Many references are old (>5 years). Recent reviews (2020–2024) on oxidative stress, NO and phytotherapeutics in depression should be included.
R-We follow the suggestion of the reviewer and incorporate new references in the discussion section.
8 - Limitations are underdeveloped, only mentioning that punicalagin/ellagic acid were not tested separately. Additional limitations should be included: use of males only (no consideration of sex differences); acute model (does not assess chronic administration); small biochemical samples (n=5); absence of behavioural markers of anhedonia (only FST/TST).
R- We have now added a Limitations section at the end of the Discussion. This section explicitly acknowledges that: (i) our conclusions regarding nitrergic involvement are indirect and based on pharmacological manipulations without direct biochemical readouts; (ii) only male mice were studied, which restricts generalizability to females; and (iii) the present study was limited to acute dosing, so the long-term efficacy and safety of P. granatum remain to be established.
9 – Conclusion: The conclusion states that pomegranate “represents a promising agent for the treatment of depression” without any caveats. The wording should be moderated, acknowledging that the results are pre-clinical in a murine model.There is no direct link back to the hypotheses. The conclusion could explicitly return to the initial hypotheses, confirming or refuting each.
R – The conclusion was rewritten accordingly.
10 - There is little guidance for future research. It only suggests studying isolated compounds. Recommendations should also include chronic studies, inclusion of females, analysis of translational doses, and clinical trials.
R – This information is now stated in the “Limitations section.”
Round 2
Reviewer 1 Report
Comments and Suggestions for Authors
I have satisfied with the responses of authors to my first report.
I have no more complains.
Reviewer 2 Report
Comments and Suggestions for Authors
The authors have appropriately revised the manuscript according to the comments made by the reviewer. I agree with acceptance of the revised manuscript in Int. J. Mol. Sc.
Reviewer 3 Report
Comments and Suggestions for Authors
Dear Authors,
I appreciate your answers and all the relevant improvements you have made to your paper.
Yours sincerely,
The reviewer.
Reviewer 4 Report
Comments and Suggestions for Authors
I consider that the authors of the article should review the citations, as there are many unnecessary self-citations.